# Using Multi-Criteria Decision Analysis to Select Waste to Energy Technology for a Mega City: The Case of Moscow

**Anna Kurbatova [1],\* and Hani Ahmed Abu-Qdais [2]** 

[1]   Department of Ecological Monitoring and Forecasting, Faculty of Ecology, Peoples' Friendship University of Russia (RUDN University), 6, Miklukho-Maklaya Street, 117198 Moscow, Russia

[2]   Civil Engineering Department, Jordan University of Science and Technology, P.O. Box 3030, Irbid 22110, Jordan; hqdais@just.edu.jo

\*   Correspondence: kurbatova-ai@rudn.ru

**Abstract:** In a mega city like Moscow, both municipal solid waste management and energy systems are managed in an unsustainable way. Therefore, utilizing the municipal solid waste to generate energy will help the city in achieving sustainability by decreasing greenhouse gases emissions and the need for land to dispose the solid waste. In this study, various Waste to Energy (WTE) options were evaluated using analytical hierarchy process (AHP) to select the most appropriate technology for the Moscow region. The developed AHP model consists of 4 levels, which assessed four WTE technologies, namely landfill biogas, anaerobic digestion, incineration, and refuse derived fuel (RDF), using four criteria and nine subcriteria. The pairwise comparison was achieved by soliciting 16 experts' opinions. The priority weights of various criteria, subcriteria, and alternatives were determined using Expert Choice Software. The developed model indicated that landfill biogas is the preferred option with a global weight of 0.448, followed by the anaerobic digestion with a weight of 0.320 and incineration with a weight of 0.138, while the least preferred technology is the RDF with a weight of 0.094. Sensitivity analysis has shown that the priorities of WTE alternatives are sensitive for the environmental and technical criteria. The developed AHP model can be used by the decision makers in Moscow in the field of WTE.

**Keywords:** waste to energy; Moscow; mega city; analytical hierarchy process (AHP); sensitivity

## 1. Introduction

According to the World Bank report, in 2017 the amount of solid waste generated globally was about 2.01 billion tones and is expected to increase to 3.40 billion tones by 2050 [1]. The report estimates that about 33% of the solid waste generated worldwide is not managed properly and instead openly dumped. Such practices are leading to several public health and environmental problems [2,3]. Furthermore, improper management of solid waste will be adversely reflected on the economic development [4]. This is especially true for Moscow City, where until recently, more than 90% of the municipal solid waste generated by the city and its suburbs (Moscow Region) is ultimately disposed of into open dumps and little fraction found its way to sanitary landfills [5–7].

One of the options to minimize the impacts of the generated solid waste is to utilize it for energy recovery through various waste to energy technologies [8]. Nowadays, more and more countries are adopting Waste to Energy (WTE) systems as a sustainable management option to solid waste generated by municipalities and industries [9]. The World Energy Council reported that the WTE market is growing steadily until the year 2023 at an annual rate of 5.5% [10]. The Confederation of

the European Waste to Energy Plants reported that in the year 2018, Germany diverted 31% of the total municipal solid waste generated in the country into WTE facilities, while during the same year in Sweden, Finland, Norway, and Denmark, the diversion reached more than 50% [11–13]. While many countries around the world are moving from a linear to circular economy, the WTE technologies became part of circular economy models [14], as shown in Figure 1 [15]. However, this is not the case in the Russian Federation, where the contribution of the renewable energy in general and the WTE in particular in the total energy mix of the country is still very low [16–18]. Realizing the importance of the waste to energy option, the draft Energy Strategy of the Russian Federation for the period up to 2035 indicates how the energy sector will be developed in the next two decades, what measures should be taken to make the energy industry more sustainable, efficient, and safe. One of the priority areas outlined in the strategy is the recovery of energy by adopting waste to energy options. [19].

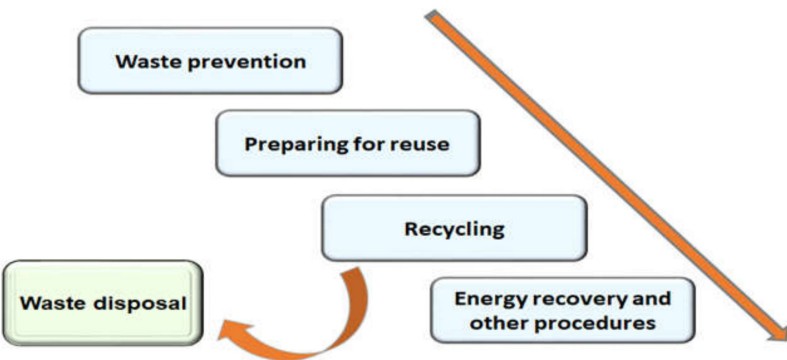

**Figure 1.** The location of waste to energy within the Circular Economy Model [15].

There are different technologies applied in energy recovery from solid waste. Each technology is capable to handle different types of waste feedstock [20]. Each WTE alternative has its own advantages and limitations. Therefore, it is of great importance to make the right decision in selecting the optimal alternative or combination of alternatives, so as to maximize the benefits from the selected options. Finding an optimal WTE technology is a challenging task that cannot be made based on a single criterion. It is rather a multivariable complex problem that needs prioritization of alternatives using a multicriteria decision making process (MCDM), according to which decision making takes place in the presence of multiple, usually conflicting, criteria [21,22]. Analytical hierarchy process (AHP) is an efficient decision making tool in cases like selection of the most suitable energy options for a certain country [23,24]. In selecting the most appropriate option of energy recovery from municipal solid waste (MSW), environmental, economic, and social aspects should be considered [25].

The main objective of this paper is to review the status of solid waste management and energy sectors in Moscow region in order to select the most appropriate waste to energy alternative to be adopted by the Russian Capital Moscow and its suburbs (Moscow Region), so as to put in the hands of decision makers a decision making tool based on which they can make their informed decisions.

### 1.1. Study Area

Moscow is the capital city of the Russian Federation. The city population in 2017 was 12 million, while the total population of the city and its suburbs (Moscow Region) is 20 million [26]. It is the largest Megacity in the northwest part of the world. The city is located at 118–255 m above the sea level and lying between latitudes 55° 54′ N and longitudes 37° 36′ E in the Central European plain between the Oka river and the Volga river (Figure 2). Moscow river is crossing the city from northwest to southeast with a length of 80 km within the city boundaries. The city is divided into 12 administrative areas that are made up of 125 administrative districts [27].

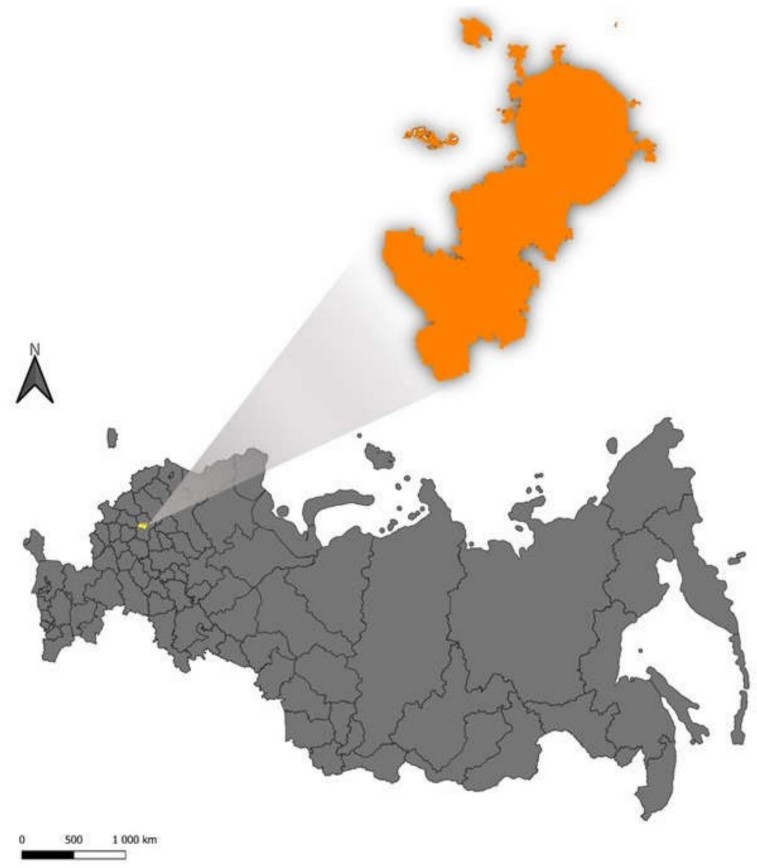

**Figure 2.** Map of Russian Federation showing Moscow region (developed by the authors).

### 1.2. Solid Waste Management in Moscow

Until recently, Moscow local authorities were responsible for the solid waste management within the Moscow region. According to the recent radical reforms that took place in the solid waste management sector of the Russian Federation, regional operators started getting involved in the solid waste management process in 2019.

In the Russian Federation, the total municipal solid waste generation has increased during the period 2013–2017 from 55.6 to 58.4 million tons [28], where 90% has been disposed in landfills and dump sites, while only 3% has been incinerated and 7% has been recycled [29,30]. On the other hand, Moscow City and its suburb (Moscow Region) annually produce 11 million tons of municipal solid waste (8 million tons from Moscow city itself, while the city suburbs generate 3 million tons annually) [31,32], which accounts for about 18.5% of the total annual MSW generated in the country. Figure 3 shows the composition of the municipal solid waste generated by Moscow [31]. More than 90% of the MSW generated amount is ultimately disposed of into open dumps and little fraction finds its way to sanitary landfills [5–7]. Landfilling of the MSW arising from the Moscow region requires 100 hectares of land annually [33]. Nowadays, there are 14 solid waste landfills surrounding the city of Moscow, and most of them are located nearby residential areas (500–900 m away) [34]. Furthermore, there are more than 100 illegal solid waste dumping sites surrounding the city, occupying a total area of 940 hectares (1% of the total city area) [27]. Such practices are unsustainable and pose risks to the public health and environment [35–37]. The generated municipal solid waste is collected in a mixed status without any sorting efforts, with a moisture content between 30–40%.

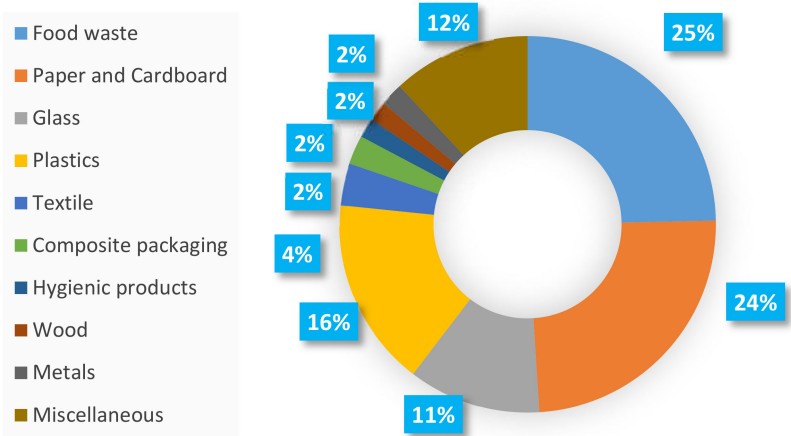

**Figure 3.** Physical composition of solid waste generated in the Moscow Region [31].

Given the fact that the solid waste management approach followed in the country is neither economical nor sustainable, the Russian Authorities have decided to follow a more sustainable and economically feasible approach. Therefore, the government has launched the Comprehensive Municipal Solid Waste Strategy in 2014. The strategy focuses on maximizing the amount of waste diverted from landfills by creating an integrated management system, which is mainly based on utilization of the generated waste through recycling and energy recovery [29].

### 1.3. Energy System and Waste to Energy in Moscow

Moscow is one of the coldest cities among megacities worldwide. As such, the city has the largest districts heating system in the world [37]. The fuel and energy system of the city is very important for sustaining the life in such a mega city. Energy consumption of the Moscow region accounts for 11.7 % of the total country energy consumption, where 97% of the energy supplied to the Moscow region is generated by different types of thermal power stations [17].

Moscow's heat demand, depending on the severity of the winter season, ranges from between 93–97 million Gcalorie per year. The total electricity generation is about 50 billion kWh, where 10 billion kWh consumed to run the power system and as losses from the network [38]. The heat and power supply of the city are provided by 13 thermal power plants and 66 district heating stations. Such a system of energy supply leads to several adverse environmental impacts. Using the fossil fuel in generating the heat and electric energy of the city leads to annual emission of about 42 million tons of carbon dioxide and about 65 million tons of water vapor to the atmosphere. Furthermore, discharges of the hot water from the thermal power plant leads to deterioration of the surface water quality in the receiving water bodies, where more than 65 million tons of low-grade hot discharges with a heat content of 110 million Gcalorie is disposed mainly into the Moscow river [38]. The heating system $CO_2$ emissions for districts emits an annual amount of 22.4 Mt $CO_2$. Such emissions can be reduced by adopting renewable energy in running the heating system of the Moscow region [39].

Although renewable energy has a great potential in a country like Russia, which accounts for 30% of the country's energy supply, the renewable energy contribution in Russia (excluding hydropower) is estimated to be not more than 0.5% [17]. This applies to the WTE, which is a relatively new concept to Russia [16,18].

In 2001, as part of the modernization of Moscow thermal power plant No. 2, three turbine generators were installed and operated using MSW to generate 1.2 MW each [16]. After that, three more plants were upgraded to incinerate MSW that generate from 3.6–12 MW of power which is relatively small [40]. In 2008, power plant No. 3 in southern Moscow had been subjected to rehabilitation and expansion to be able to incinerate part of unsorted municipal solid waste arising from Moscow for energy recovery. In 2017, the plant incinerated 309.6 thousand tons of municipal solid waste from

Moscow. As a result, 55.4 thousand MWh of electrical energy has been generated. Furthermore, 376.2 thousand Gcalorie of heat energy has been generated, most of which has been supplied to the electrification and heating purposes of Moscow. [41].

As part of Clean Country Project, recently, an international consortium including a Russian company has been awarded a contract to build four modern waste to energy plants in the Moscow region. The plants will annually treat 2.8 million tons of municipal solid waste (each 700 thousand tons/year) that arise from 5 million inhabitants in the Moscow region. The four plants are expected to generate 280 MW of power that will deliver the electricity to 1.5 million people and will reduce the amount of landfilled waste by 25% in the Moscow region [18].

## 2. Methodology

### 2.1. Identification of the Goal and Criteria

Since the main objective of the study is to make a decision regarding the appropriate waste to energy option for Moscow, the AHP model goal has been formulated to reflect such an objective. To choose the criteria that will be used in the alternative evaluation, a comprehensive literature review was conducted on the waste to energy in general and on solid waste and energy sectors of Russia. After that, a group of 8 criteria was listed which then reduced to 3 criteria that are most relevant to such a region like Moscow.

### 2.2. AHP Model Construction

The AHP model in this study consists of four levels. The first level presents the goal of the study problem, which is selecting an appropriate waste to energy option for Moscow. The second level is the selection criteria of the waste to energy option which consists of three main criteria, namely environmental, technical, and socioeconomic criterion. Furthermore, level three of the hierarchy is devoted to the sub-criteria, where each of the three main criteria involves three sub-criteria. Table 1 shows a description of the criteria and subcriteria that were used in the development of the AHP model.

**Table 1.** Description of criteria and subcriteria in the analytical hierarchy process (AHP) model.

| Main Criteria | Subcriteria | Description |
|---|---|---|
| **Environmental and Health** | *Public and occupational health (POH)* | Capability of the selected technology to minimize the risks on public and workers' health |
| | *Pollution potential (PP)* | Minimal adverse environmental impacts on water, soil and air |
| | *Climate change impact (CCI)* | Carbon footprint of the selected technology that has less emissions of carbon dioxide and other greenhouse gases |
| **Technical** | *Energy production (EP)* | The selected technology with the highest energetic potential |
| | *Availability of know how (AOH)* | The existence of practical knowledge and skills regarding running and maintaining the selected waste to energy technology |
| | *Sophistication of technology (SOT)* | Refers to high and advanced technology that requires skillful human resources |
| **Socioeconomic** | *Capital cost (CC)* | The selected waste to energy technology that has the least initial investment cost (CAPEX) |
| | *Operation and maintenance cost (OMC)* | The selected waste to energy technology that has the least running and maintenance costs (OPEX) |
| | *Job creation (JC)* | The potential of the selected technology to create employment opportunity |

Finally, alternatives that will be considered for evaluation to achieve the study goal are presented under the fourth level of the AHP model, which include landfill biogas plant, anaerobic digestion, incineration, and refuse derived fuel (RDF). Table 2 presents a description of the four waste to energy alternatives considered by the study, while Figure 4 shows the structure of the AHP model.

**Table 2.** Description of waste to energy alternatives used in AHP analysis.

| Waste to Energy Alternative | Description |
|---|---|
| **Landfill gas (LFG)** | A sanitary landfill with a gas plant erected on the landfill to recover the gas generated as a result of anaerobic degradation of the organic fraction to produce heat or electric energy. |
| **Anaerobic digestion (AD)** | Forwarding the biodegradable organic fraction of the municipal solid waste to a specially constructed plant where the waste is subjected to anaerobic degradation and producing methane gas for energy production. The digestate and residue after digestion need to be disposed. |
| **Incineration plant (IP)** | A thermal treatment process, where the solid waste is combusted in a solids chamber at high temperature. The heat is recovered from the system where it is used for heating purposes or electricity generation. While the gases are burned in a secondary chamber. The emitted gases may contain harmful emissions while the ash requires disposal. |
| **Refuse derived fuel (RDF)** | Combustible fraction of municipal solid waste including paper, plastic, organics, and wood are densified into pellets that can be utilized by different industries for energy production. |

### 2.3. Pairwise Comparison by Solicitation of Experts Opinions

In AHP analysis, expert opinion and judgement is an important step in the decision making process [42]. To solicit the experts' opinion, a special questionnaire was prepared. The questionnaire including guidelines for the interviewees on how to carry out the comparison, as well as matrices to conduct the pairwise comparison based on Saaty's scale of 9 points The comparisons are made using a scale of absolute judgements that represents how much more one element dominates another with respect to a given attribute (Table 3) [43,44].

**Table 3.** Scale for pairwise comparison (Saaty, 1980).

| Importance Scale | Definition of the Importance Scale |
|---|---|
| 1 | Equal importance of the row criterion over the column criterion |
| 2 | Between equal and weak importance of the row criterion over the column criterion |
| 3 | Weak importance of the row criterion over the column criterion |
| 4 | Between weak and strong importance of the row criterion over the column criterion |
| 5 | Strong importance of the row criterion over the column criterion |
| 6 | Between strong and demonstrated importance of the row criterion over the column criterion |
| 7 | Demonstrated importance of the row criterion over the column criterion |
| 8 | Between demonstrated and absolute importance of the row criterion over the column criterion |
| 9 | Absolute importance of the row criterion over the column criterion |

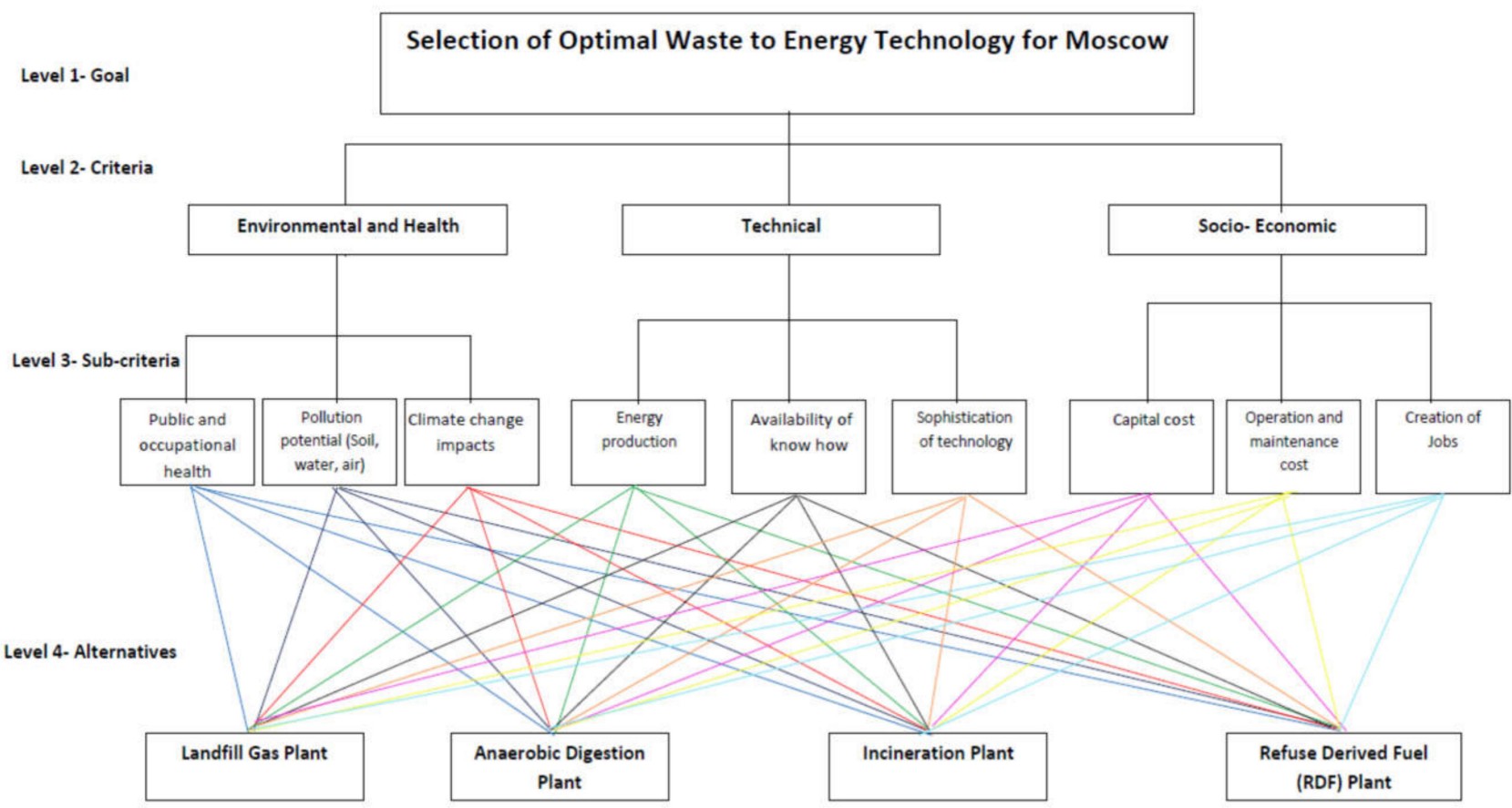

**Figure 4.** Analytical Hierarchy Model for the selection of Waste to Energy Technology Energy Technology.

Sixteen experts and stakeholders with diverse backgrounds who are familiar with the waste and energy sectors of the Moscow region were interviewed and consulted by conducting brainstorming sessions and work groups. The initial screening of the opinions revealed in the rejection of three interviews due to incompleteness and inconsistency in the information provided. As such, 13 experts' opinions were considered in the study. Table 4 shows the numbers and categories of the experts that took part in filling the questionnaire. In each matrix, the experts' judgment was reflected by giving a weight to each per criteria and sub-criteria that were subjected to compare towards achieving the study goal. From the pairwise comparison of various criteria and subcriteria to achieve the study goal, square matrices were derived. The score values of pairwise comparison expressed by the experts in each pairwise comparison matrices were aggregated using additive AHP procedure (by calculating the arithmetic mean for matrices developed by the experts) [45] and subjected to normalization, after which the Eigenvector were calculated and used to weight the results for the four waste to energy alternatives [14].

**Table 4.** List of the consulted experts' categories, their professions, and number.

| No. | Experts Category | Profession | Number |
|---|---|---|---|
| 1. | Academics | Lecturers and researchers with environmental background | 4 |
| 2. | Waste Professionals | Solid waste treatment plants operators | 2 |
| 3. | Decision Makers of Federal and Local Authorities | Specialist of the natural resources protection and management | 3 |
| 4. | Energy Specialist | Operators and researchers in the field of energy | 3 |
| 5. | Members of International donating agencies in Moscow | Head of Department | 1 |
| 6. | Graduate Researchers | PhD researchers in integrated solid waste management | 3 |

To ensure the consistency of the experts' judgments, a consistency check was performed for each matrix by calculating the consistency ratio (CR). First the eigenvalue ($\lambda_{max}$) was calculated according to Equation (1).

$$A \cdot w = \lambda_{max} \cdot w \tag{1}$$

where A is the comparison matrix, w is the normalized eigenvector (priority vector), and $\lambda_{max}$ is the eigenvalue. After that, the consistency index has been calculated using Equation (2).

$$CI = \frac{l_{max} - n}{n - 1} \tag{2}$$

Considering the randomness in judgment, the consistency ratio by Equation (3):

$$CR = \frac{CI}{RI} \tag{3}$$

where RI is the random index which expresses expected value of the CI corresponding to the order of matrices. Table 5 shows the values of the random index.

**Table 5.** Random index (RI) values for different matrix sizes.

| n | 1 | 2 | 3 | 4 | 5 | 6 | 7 | 8 | 9 |
|---|---|---|---|---|---|---|---|---|---|
| RC | 0 | 0 | 0.58 | 0.90 | 1.12 | 1.24 | 1.32 | 1.41 | 1.45 |

In the case where the CR value is within acceptable range (usually less than 10%), the judgments are considered consistent.

*2.4. Priorities Assessment*

After the judgement's consistency was checked, synthesizing the judgments by aggregating the weights through hierarchy was performed to determine the combined priorities of each waste to energy alternative for the Moscow region. To achieve that, expert choice software was used.

## 3. Results and Discussion

Since the renewable energy market is not yet well established in Russia [46], the waste to energy concept is still in its infancy [16,18]. However, the recent development in the Russian regulatory framework in the field of renewable energy and solid waste are paving the road towards the development and establishment of the waste to energy system in the country [40]. The question remaining, which waste to energy technology is most suiting the local circumstance in Russia in general and in Moscow in particular, which the current study is trying to answer.

*3.1. Application of AHP*

AHP is a widely used tool for decision making processes in the field of waste management. By reviewing 279 articles published between 1980 and 2015, Coelho et al. (2017) [47] reported that AHP was the dominant decision making process used among other processes with 32% (91 articles) of the total. Few researchers have used AHP in the field of energy studies in Russia. For example, Schlifter and Madlener (2016) [48] used AHP in analyzing the risks associated with energy performance contracting projects in Russia. Another study conducted by Geller (2018) [49] assessed the energy security in Russia using the AHP approach, while Zaychenko et al. (2018) [50] used AHP to adjust the energy strategy of Russia to suit the Far North part of the country. Vershinina et al. (2020) used AHP among other multicriteria methods in efficiency analysis of using waste-based fuel mixtures in the power industries of China, Japan, and Russia [51]. Up to the knowledge of the authors, no studies were conducted to assess the waste to energy options in Russia using the AHP approach.

At the beginning, the main criteria (environmental-health, technical, and socioeconomic) were subjected to pairwise comparison with respect to the goal. The pairwise comparison matrix (Table 6) shows the weights assigned for each criterion by the experts and its calculated priority. The calculated consistency ratio was found to be 0.08, which is acceptable since it is less than 0.1. It can be observed that the pairwise comparison revealed that environmental criterion has a demonstrated strong importance with respect to technical as well as with respect to socioeconomic criteria. On the other hand, pairwise comparison showed that technical criterion is almost of equal to weak importance as compared to socioeconomic criterion.

**Table 6.** Pairwise comparison matrix for assessing relative importance of main criteria with respect to the main goal.

| Criteria | Environmental and Health | Technical | Socioeconomic | Priority Vector |
|---|---|---|---|---|
| **Environmental and Health** | 1 | 6 | 5 | 0.729 |
| **Technical** | 0.167 | 1 | 2 | 0.162 |
| **Socioeconomic** | 0.20 | 0.5 | 1 | 0.109 |

Figure 5 shows the priorities of the main criteria with respect to the goal. It can be seen that the environmental and health criterion has the highest weight of 0.729, followed by the technical criterion with a weight of 0.162, while the socioeconomic criterion has the least weight of 0.109. This indicates that the main concern of the respondents in selecting the most sustainable alternative for energy

recovery from waste is the environment. This is may be attributed to the fact that solid waste generated in Moscow was until recently still managed in a traditional way that follows the "cradle to grave" approach. Such practices are associated with environmental and public health impacts that are reflected adversely on the citizens' health and their environment [35–37]. Furthermore, Russian energy sector also poses adverse environmental impacts where the fossil fuel (oil and gas) is responsible for 76.1% of the methane emissions in Russia [52]. Similar findings were indicated by other researchers such as Qazi et al. (2018) and Khoshand et al. (2018) [25,53] who reported that environmental criterion ranked first when considering the waste to energy options in Oman and Iran, respectively.

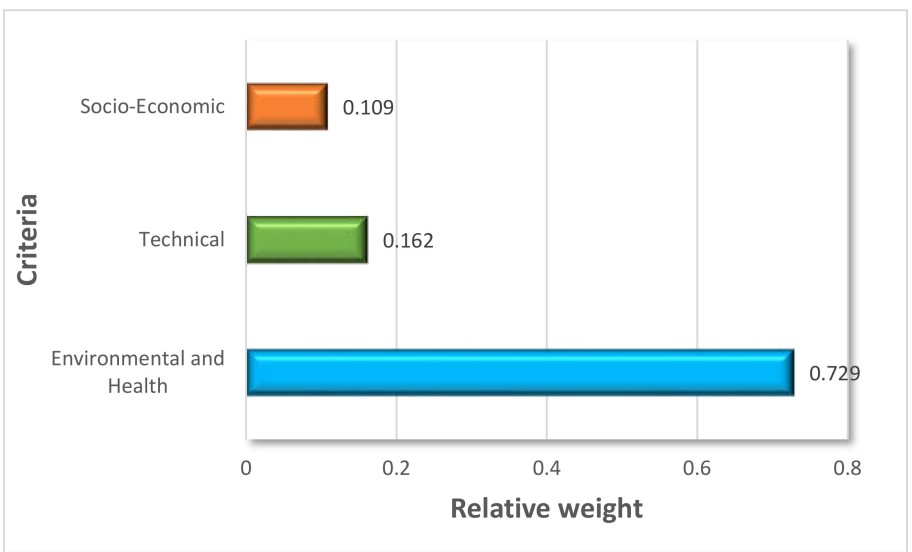

**Figure 5.** Priorities of the main criteria with respect to the goal.

To prioritize the nine considered subcriteria to select the most appropriate option of the waste to energy alternatives for Moscow, three pairwise comparison matrices with respect to each main criterion were developed based on the expert opinion. Table 7 summarizes the pairwise comparison of the subcriteria with respect to main criteria. It can be seen from the table that under environmental and health criterion, the public and occupational health (POH) has received the highest rank of 0.683 as compared to the other two subcriteria. The high weight assigned to POH subcriteria as compared to the other subcriteria may be explained by the fact that soliciting of the experts' opinion took place during the COVID-19 pandemic spread out in Moscow, where the public health issue was a priority for everybody, which is expected to be reflected on the solicited opinion by ranking the POH. Khoshand et al. (2018) [25] also found that the occupational health and safety has the highest priority as compared to other criteria.

Under the technical criteria, the potential of energy production received the highest rank of 0.709 as shown from Table 7, while the availability of knowhow and sophistication of technology subcriteria received 0.178 and 0.113 relative weights, respectively. This indicates that regardless of the technology used and its complexity, the main issue in selecting that technology is the amount of the generated energy from that WTE technology. Qazi et al. (2018) [53] found that energy production is also of the highest priority.

**Table 7.** Pairwise comparison of subcriteria with respect to main criteria.

| Criteria | Environmental- Health | | | Technical | | | Socioeconomic | | | Priority Vector |
|---|---|---|---|---|---|---|---|---|---|---|
| Subcriteria | POH | PP | CCI | EP | AKH | SOT | CC | OMC | JC | |
| POH | 1 | 4 | 5 | - | - | - | - | - | - | 0.683 |
| PP | 0.25 | 1 | 2 | - | - | - | - | - | - | 0.20 |
| CCI | 0.20 | 0.50 | 1 | - | - | - | - | - | - | 0.117 |
| EP | - | - | - | 1 | 5 | 5 | - | - | - | 0.709 |
| AKH | - | - | - | 0.20 | 1 | 2 | - | - | - | 0.178 |
| SOT | - | - | - | 0.20 | 0.50 | 1 | - | - | - | 0.113 |
| CC | - | - | - | - | - | - | 1 | 3 | 4 | 0.615 |
| O&MC | - | - | - | - | - | - | 0.33 | 1 | 3 | 0.268 |
| JC | - | - | - | - | - | - | 0.25 | 0.33 | 1 | 0.117 |

POC = Public and occupational health, PP = Pollution potential, CCI = Climate change impact, EP = Energy production, AKH = Availability of knowhow, SOT = Sophistication of Technology, CC = Capital cost, OMC = Operation and maintenance cost, JC = Job creation.

On the other hand, when considering the subcriteria under the socioeconomic criterion, the capital cost (CC) ranked first with a weight of 0.615, followed by the operation and maintenance cost (OMC) with a weight of 0.268, while the least priority was given to job creation (JC) with a weight of 0.117. According to the experts' opinion, the initial investment cost of the technology is more important than the operation and maintenance cost and the number of employment opportunities that may be created as a result of adopting WTE technology in Moscow. A similar finding was reported by Rahman et al. (2017) [20], who found that the plant establishment cost has gained the highest priority. Although job creation is one of the important aspects in considering various types of waste management projects, it has received the lowest priority among the socioeconomic subcriteria in Moscow. This is may be due to the relatively low unemployment level in Russia, which is less than 5% [54], where the creation of employment opportunities is not that important as compared to capital, operation, and maintenance costs.

The next step in the pairwise comparison involved the assessment of the WTE alternatives based on the subcriteria. The judgment of the experts during the comparison revealed 9 matrices that resulted in priority vectors. Figure 6 shows the ranking of the considered technologies based on the subcriteria. Considering the POH criteria, landfill gas from a sanitary landfill is considered the best alternative, followed by anaerobic digestion as a second priority, while incineration and refuse derived fuel are the least preferred alternatives. It seems that experts prefer the biochemical technologies like landfill biogas and anaerobic digestion over the thermal ones (i.e., incineration and RDF). A similar trend in preferences is observed under pollution potential and climate change impacts subcriteria, where the anaerobic digestion ranked first followed by landfill gas and the least preferred options were incinerator and RDF with almost similar weights. Fan et al. (2019) [55] assessed the GHG emissions ($CO_2$eq) of incineration and anaerobic digestion (AD) under different electricity production mix for different countries. The study estimated that for Russian Federation, emissions from incineration of MSW is 105.54 kg of $CO_2$ equivalent per ton of solid waste incinerated, while it is 70.22 kg of $CO_2$ equivalent per ton of solid waste subjected to anaerobic digestion (AD), which implies biochemical processes like anaerobic digestion emitting 33% less greenhouse gases than incinerators. Compared to landfills, incinerators emissions are also higher, where sanitary landfills emit 50% less greenhouse gas emissions than incinerators [56].

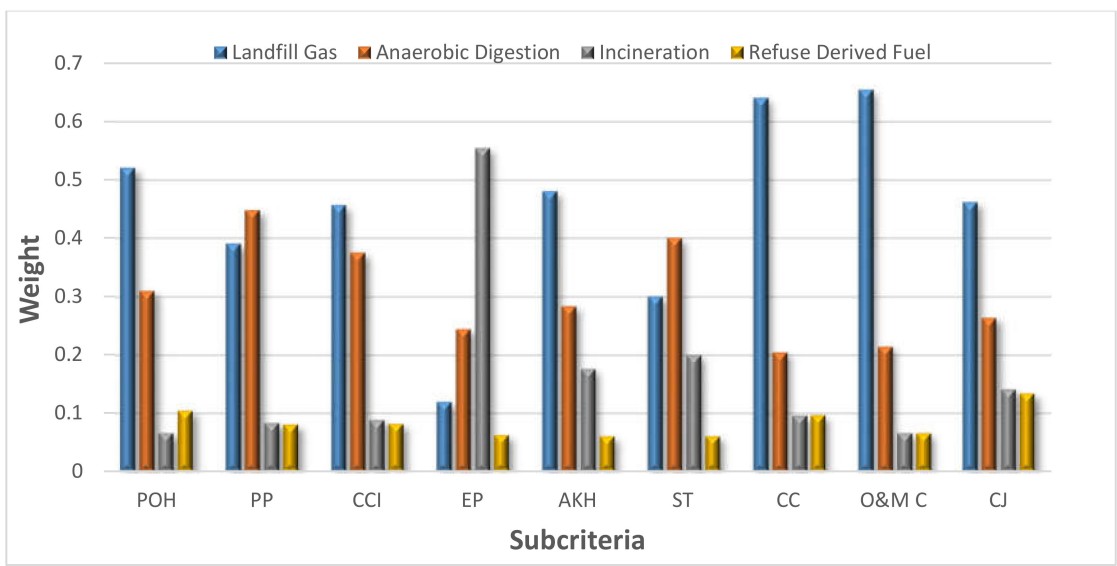

**Figure 6.** Ranking of waste to energy technologies based on the considered subcriteria.

Based on the potential energetic production (EP), subcriteria incineration technology ranked first with a weight of 0.554, followed by anaerobic digestion with a weight of 0.245. The landfill gas ranked third option with a weight of 0.119, while the RDF was the least preferred with a weight of 0.062. This finding is not surprising in a cold region like Moscow, where the energy generated from incinerators can be used for district heating which demands between 93–97 million Gcalorie per year [38]. On the basis of per ton of solid waste treated, incineration energy production was found to be about 13 times (2.34 GJ/ton) greater than landfill gas and 5 times greater than anaerobic digestion [57]. As for Moscow, the four incineration plants currently under construction will treat an amount of 2.8 million tons of municipal solid waste arising from the Moscow region; the potential annual power production of the four incineration plants is 280 MW [18]. This will generate an electric energy of 3.15 GJ/ton (0.876 MWh/ton) of solid waste treated. On the other hand, the potential power that can be generated from the already landfilled solid waste in Moscow landfills is 901 MW [58].

Under the availability of knowhow subcriteria, the landfill gas alternative received the highest rank, followed by anaerobic digestion, while the incineration and RDF received the least preference, as they are relatively new technologies to Russia [16] where is no sufficient expertise to run such plants. The same order of waste to energy ranking is applied under all socioeconomic subcriteria. The highest preference is being given to landfill gas, followed by anaerobic digestion, while the thermal processes of incineration and RDF were the least preferred as presented in Figure 6.

The final step in the AHP analysis is to determine the global priorities of WTE alternatives. This can be achieved by synthesizing the local priorities of all criteria, subcriteria, and alternatives to obtain the global weight for each alternative. To determine, the local priorities across all matrices of criteria are synthesized by multiplying the local priority vector of alternatives by local priority vector of each criterion and aggregated to get the final priority vector (global weight for each alternative) through additives aggregation with normalization of the local criteria priorities to unity as follows [25,59]. Figure 7 shows the global (overall) priorities of the waste to energy technologies for the Moscow region. It can be observed that landfill biogas is the best alternative of waste to energy with a global weight of 0.448. Landfill biogas recovery can be applied to the existing landfills in the Moscow region, where such a technology is not being practiced at a reasonable scale. Anaerobic digestion of solid waste ranked as a second preferred technology with a weight of 0.32. The thermal processes of incineration and refuse derived fuel are the least preferred technologies with weights of 0.138 and 0.094, respectively.

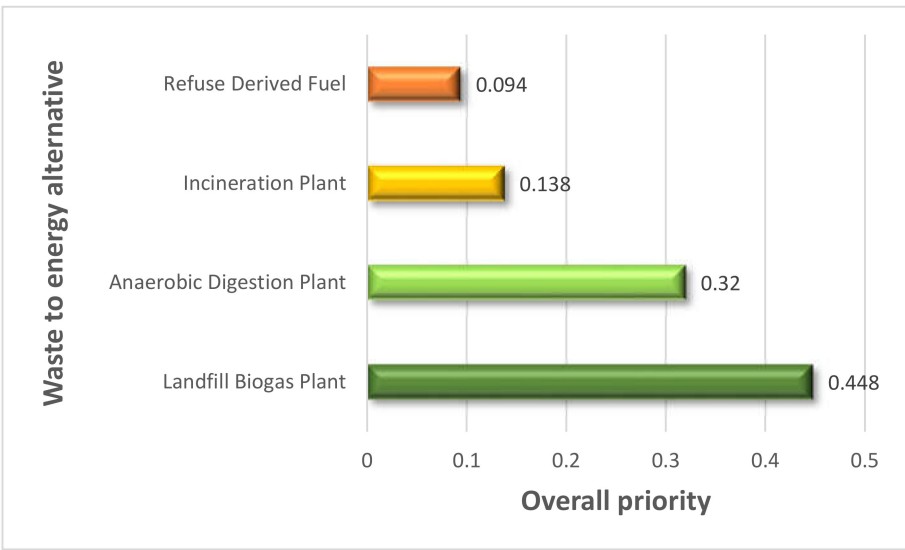

**Figure 7.** Global priorities of the waste to energy alternatives for Moscow region.

Although incineration has a potential to produce energy more than other technologies of landfill gas and anaerobic digestion, the incineration option has been ranked as the third preferred one. The reason behind that is the solid waste incineration still facing strong public opposition in Russia due to perceptions that are relevant to human health risks. For example, as a result of a claim from environmentalists, in early 2019, the Supreme Court of the Russian Republic of Tatarstan stopped the construction of an incinerator and a landfill to dump the residue slag and ash from such incinerator [60]. This suggests that the incinerator technology promoters in Russia need to encourage using state of the art incineration technology with a minimal pollution potential and to disseminate the information to the public in Russia in a transparent manner. Furthermore, adopting effective sorting of the solid waste at the generation source will increase the calorific value of the waste that will be incinerated, which renders the incineration as a more attractive technology [61]. Therefore, it is of great importance to move as soon as possible toward a source separation in the Moscow region. It should be noted that countries should not depend heavily on one WTE, as this may affect the circular economy targets [13]. For example, as a result of the excessive dependence on incineration, there is a doubt that Nordic countries can meet the EU recycling targets [62]. Madrid Municipality is another example, where it plans to phase out the incineration in the capital city.

*3.2. Sensitivity Analysis*

The AHP results are mainly dependent on the priorities indicated by the experts during the pairwise comparison process. As such, any change in the weights of the criteria and subcriteria may affect the rank of the WTE alternatives. To assess the sensitivity of alternatives ranking, dynamic sensitivity analysis was conducted based on different scenarios [25] using Expert Choice Software [42].

The following scenarios were used in the sensitivity assessment:

1. All criteria have an equal weight of 33.3%
2. One criteria having 100% weight while the other two having weights of 0%.

Figure 8 shows an example of the sensitivity alternatives based on the first scenario of having equal weights of criteria. It can be seen that the incineration technology has become the first preferred WTE option followed by landfill biogas technology with a weight close to each other. The third option is anaerobic digestion, while the least preferred one is the RDF. Table 8 shows the ranking of the various technologies based on both scenarios. As it can be seen, when the environmental and health criteria had a weight of 100%, while other criteria had 0% (Scenario 2.a), the order of the technologies remains

the same but with different weights. However, under scenario 2.b with 100% of technical criteria, the incineration option moves to rank as a first option followed by anaerobic digestion as a second one, while landfill biogas and RDF rank third and fourth options, respectively. Finally, under scenario 2.c, where the socioeconomic criterion has a weight of 100%, the order of alternatives did not change.

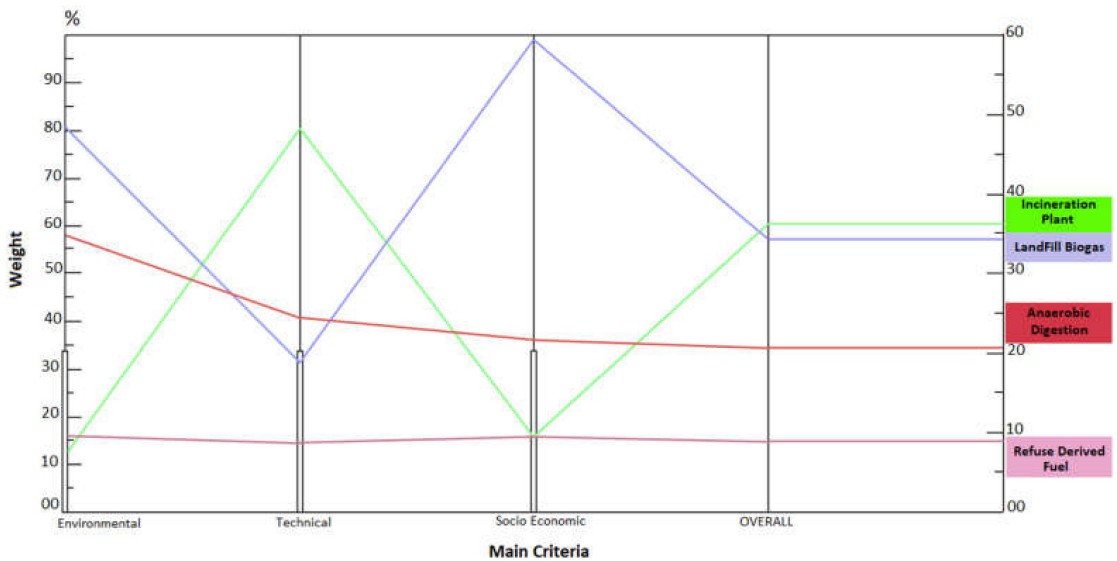

**Figure 8.** Sensitivity analysis under equal weight of main criteria.

**Table 8.** Rankings of waste to energy alternatives based on sensitivity analysis.

| Scenario | Criteria | Criteria Weight | Alternative | Alternative Ranking |
|---|---|---|---|---|
| **Scenario No. 1** | - Environmental and Health<br>- Technical<br>- Socioeconomic | 33.33%<br>33.33%<br>33.33% | - Landfill biogas<br>- Anaerobic digestion<br>- Incineration<br>- Refuse derived fuel | 2<br>3<br>1<br>4 |
| **Scenario No. 2. a** | - Environmental and Health<br>- Technical<br>- Socioeconomic | 100%<br>0%<br>0% | - Landfill biogas<br>- Anaerobic digestion<br>- Incineration<br>- Refuse derived fuel | 1<br>2<br>3<br>4 |
| **Scenario No. 2. b** | - Environmental and Health<br>- Technical<br>- Socioeconomic | 0%<br>100%<br>0% | - Landfill biogas<br>- Anaerobic digestion<br>- Incineration<br>- Refuse derived fuel | 3<br>2<br>1<br>4 |
| **Scenario No. 2. c** | - Environmental and Health<br>- Technical<br>- Socioeconomic | 0%<br>0%<br>100% | - Landfill biogas<br>- Anaerobic digestion<br>- Incineration<br>- Refuse derived fuel | 1<br>2<br>3<br>4 |

## 4. Conclusions and Recommendations

Selection of the right WTE technology is a complex decision process and it involves many factors and variables. This is especially true for Moscow, which is the largest Megacity in the northwest part of the world, where the solid waste and energy should be managed in a more sustainable manner. In this study, a detailed environmental, technical, and socioeconomic assessment was carried out to select the appropriate WTE option for the Moscow region. AHP analysis was used to make a decision among four WTE alternatives, which treat the whole stream of solid waste or some of its components. A four level hierarchy model was developed and used by soliciting experts' opinions.

Experts with a diverse background participated in the pairwise comparison process based on a specially designed questionnaire.

Based on different criteria and subcriteria, the considered WTE technologies were ranked. The ranking results show that the landfill biogas is the most preferred waste to energy technology for Moscow, followed by anaerobic digestion technology. Despite the fact that Moscow soon will operate large incinerators, incineration option was ranked as the third preferred technology. This is attributed to the public perceptions on the pollution potential of incineration technology, and the sophisticated technology which requires the availability of knowhow and expertise to operate such thermal processes including the RDF. The sensitivity analysis showed that incineration option ranks as the most preferred one when the main criteria had equal weights.

Currently, there are several WTE technologies available that have proved their feasibility in different regions of the world. Therefore, each country should assess and select the suitable technology that suits its local circumstances. AHP as a multivariate decision making tool can help in achieving such objective.

**Author Contributions:** A.K. collected the literature and data, prepared the questionnaire and conducted the interview with the experts to solicit their opinion, and participated in the text writing up. H.A.A.-Q. interpreted the data, analyzed the results of the pair wise comparisons after which he fed the results into the software and prepared the manuscript text. All authors have read and agreed to the published version of the manuscript.

**Funding:** This research received no external funding.

**Acknowledgments:** The publication has been prepared with the support of the "RUDN University Program 5-100".

**Conflicts of Interest:** The authors declare no conflict of interest.

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
