# Peer review of "Using Multi-Criteria Decision Analysis to Select Waste to Energy Technology for a Mega City: The Case of Moscow"

_sustainability, doi:10.3390/su12239828_

Round 1

Reviewer 1 Report

Table 2 especially needs to be revised:

A sanitary landfill with a landfill gas to power generation option ( landfills generate landfill gas NOT biogas because landfill gas other impurities I.e. siloxanes) Landfill in NOT biogas

Anaerobic digestion generates biogas not Methane! That methane coming from organic waste is considered as "Renewable Natural Gas"

Both landfill gas and biogas contain methane if they are treated and  upgraded

so authors should understand the differences between landfill gas, biogas and "renewable natural gas" 

Also Landfill gas to energy may be preferable in Russia however landfill gas recovery is not 100% Landfills at least leak into the atmosphere 15-20% which 65% of the gas is methane. Landfills do contribute to climate change. Therefore in the West we are trying to divert organics from landfills. I do not also agree with the statement of " There is no WTE technology or group of technologies that suits all countries"   All the countries should see their practices and make informed decisions. Circular carbon economy concept should be considered. First anaerobic digestion can capture at least 95% of biogas and digestate part  can also be used for displacing fossil and mineral based fertilizers. 

Author Response

Reviewer 1

Comments and Suggestions for Authors

  1. Table 2 especially needs to be revised:

A sanitary landfill with a landfill gas to power generation option ( landfills generate landfill gas NOT biogas because landfill gas other impurities I.e. siloxanes) Landfill in NOT biogas

Anaerobic digestion generates biogas not Methane! That methane coming from organic waste is considered as "Renewable Natural Gas"

Both landfill gas and biogas contain methane if they are treated and  upgraded

so authors should understand the differences between landfill gas, biogas and "renewable natural gas" 

Agree with the reviewer, table 2 has been revised to address the reviewer's concern. Please see the changes in table 2 with red font.

  1. Also Landfill gas to energy may be preferable in Russia however landfill gas recovery is not 100% Landfills at least leak into the atmosphere 15-20% which 65% of the gas is methane. Landfills do contribute to climate change. Therefore in the West we are trying to divert organics from landfills.

Totally agree with the reviewer, due to the leak and migration of the gas, landfill gas capture in a very well designed landfills barely exceeding 60% of the gas generated. However, based on the current situation of the Russian solid waste sector, which is to date a disposal driven sector, where more than 90% of the solid waste including organic fraction finds its way to landfills with gaseous emissions being released to the atmosphere, the landfill gas recovery seems to be an attractive option as a waste to energy alternative as it will eliminate such a serious environmental problem of emitted gases.

  1. I do not also agree with the statement of " There is no WTE technology or group of technologies that suits all countries"   All the countries should see their practices and make informed decisions. Circular carbon economy concept should be considered. First anaerobic digestion can capture at least 95% of biogas and digestate part can also be used for displacing fossil and mineral based fertilizers. 

This statement may be misunderstood by the reviewer, what is meant here is that not all WTE technologies suiting all countries. Otherwise no need for such research like the current one that uses decision making tool in selecting WTE options for Moscow. What the authors wanted to say here is that each country has to select WTE technologies based on the local conditions and circumstances. To eliminate such misunderstanding, the statement has been changed to be read as follows as shown in the conclusions section:

Currently, there are several WTE technologies available that have proofed their feasibility in different regions of the world.

Reviewer 2 Report

It is a well written manuscript that falls within the scope of the “Sustainability” Journal. However, several issues need to be addressed before this manuscript is ready for publication.

Insert “Introduction heading”. Separate this section from the Abstract.

Enhance the abstract. Basically include the background, the objectives, the methodology used (there is no mention to this now), the results, and the main conclusions.

Table 4: “Academicians”; is this really what the authors want to include in this field? Maybe the term “academics” would be more appropriate. The term “academicians” is somewhat wider and is normally used in another context.

Better split the results and discussion (not discussionS) in two separate sections.  This would make the manuscript more reader friendly.

The figure on page 9, apart from having no caption and not be introduced in the main body of the manuscript, is very sloppy, the edges of the graph are everywhere and boxes are cut in half, making the figure chaotic. Please revise.

Figure 8 is so blurry, that the labels cannot be read.

Author Response

Reviewer 2

Comments and Suggestions for Authors

It is a well written manuscript that falls within the scope of the “Sustainability” Journal. However, several issues need to be addressed before this manuscript is ready for publication.

  1. Insert “Introduction heading”. Separate this section from the Abstract.

Introduction heading has been inserted, separated from the abstract and given section number 1

  1. Enhance the abstract. Basically include the background, the objectives, the methodology used (there is no mention to this now), the results, and the main conclusions.

The abstract has been rewritten to comply with the journal instructions for author. According to the instructions, the abstract should be a one paragraph with word count not exceeding 200 words. Considering such a limitation, the authors tried to have an abstract that reflects the background, main objectives, methodology and main results and conclusions.

  1. Table 4: “Academicians”; is this really what the authors want to include in this field? Maybe the term “academics” would be more appropriate. The term “academicians” is somewhat wider and is normally used in another context.

Agree, academics is more specific and refers to teachers, faculty and researchers in a university. As such academicians has been changed with academics.

  1. Better split the results and discussion (not discussionS) in two separate sections.  This would make the manuscript more reader friendly.

The nature of this article where there are main criteria, subcriteria and alternatives needs to discuss the findings in conjunction with results. The authors believe that splitting the results from discussion will dilute the concepts and will make it difficult to follow the findings. Therefore, we appreciate accepting to keep them together in one section.

The word Discussions in the section title has been changed to Discussion as per the reviewer request.

  1. The figure on page 9, apart from having no caption and not be introduced in the main body of the manuscript, is very sloppy, the edges of the graph are everywhere and boxes are cut in half, making the figure chaotic. Please revise.

The figure on page 9 (Figure 4) has been revised to become more clear as per the reviewer request. Caption has been added as well.

  1. Figure 8 is so blurry, that the labels cannot be read.

Figure 8 has been revised to have a readable label as per the reviewer request.

Reviewer 3 Report

General comments:

This study reports on evaluation of different waste to energy options in the Moscow region, using multi criteria analytical hierarchy process.

The manuscript is not really well written, however, the research is interesting. The structure follows the required structure, but the manuscript contains dozens of spelling and stylistic errors as well as shortcomings connected to the format and editing details required in the authors’ guide. Almost all parts of the manuscript should be improved. Furthermore, I have several technical questions regarding the details, which could be clarified and many supplements should be done. In my opinion, the manuscript's content is oversimplified regarding the WTE technologies and extensive English editing in needed

I believe that the four energy production methods are too few to make an informed decision, and that the individual energy production options are almost completely different, and in many cases it would be advisable to build systems containing two or more of them. Within the four main solutions, it would be necessary to share additional types (subcategories) to select the appropriate technology.

All in all, I suggest rejecting the manuscript due to the shortcomings of it. On the other hand, the author’s research is valuable, that is why I encourage the Authors to resubmit the article after modifications. In order to further develop it, I make the following recommendations.  

Note: I think, although this scientific method is also useful, an informed decision could be made using other analyzes and methods. I know that multi-criteria decision analysis is a commonly used method, but in my opinion it is quite subjective and theoretical. E.g. feasibility studies, scenario analyzes and LCA could be more useful in order to make an informed decision. Along with this, the analysis carried out is interesting, however, in my view it is less applicable to sound decision-making.

Technical recommendations:

- Probably the official template was not used during editing the manuscript. There are lots of shortcomings and inaccuracies regarding the format (no line numbers, that is why I can not indicate the exact line numbers in my review report; different format in the Abstract, tables, references etc.).

- Abstract: Landfill biogas: unnecessary capital letter.

- Page 2 (P2) Paragraph 1 (P1) (P2P1): World Bank report: this citation (2011) is too old, out-of-date. Please replace this and similar references and data with up-to-date sources and data.; tones: spelling error.

- P2P1: No dot at the end of the sentence.

- P2P2: WTE: Please insert the abbreviation only at the 1st appearance.

- unnecessary parentheses (e.g. P2P2 reference 14, 18)

- Figure 1: the given citation could be indicated below the title of the figure in a separate row.

- What does the curved arrow mean between Recycling and Waste disposal? Why is it there, not below the last option?

- P2 last paragraph: Finding…criterion: This sentence is a bit too simple wording in such a scientific article, please rewrite it.

- P2 last paragraph: decisions: is plural needed?

- P3P1: What is MSW? Municipal solid waste? Please indicate. This abbreviation is not yet explained.

- Part 1.1. first sentence: unnecessary spaces.

- P3 part 1.1. please unify the writing of million (capital is not needed).

- dot is unnecessary before reference 27.

- Figure 2.: is it own construction? If not, please add the source here and elsewhere to other figures and tables.

- P3 part 1.2.: missing dot at the end of the sentence and missing comma after ref. 28. Please check the whole document for missing characters.

- P4. ref. [5.6.7]: the format is not as required.

- more than 500 m, but not more than 900 m from the residential areas? (500-900 meters away)

- P4 P2: duplication of words: that that, please remove one of them.

- P4P2: I recommend to use "based" instead of "depending".

- Figure 3.: - unnecessary "h" in the word Hhygienic; and I recommend to use "in" instead of "by"

- I think GWh could be a better measure than Gcal, but the change is only an option.

- P4 last paragraph, 42 million tons and 65 million: Is it an own calculation or not? Maybe from ref. 38.? What is the measure of the second one?

- P5P1: The heating system's CO2 emissions"...; and Mt is for million or metric? Please clarify.

- P5P2: "This ..., in particular": the values in this sentence is completely contradictory, please modify.

- Please unify the use of Gcalorie in the text.

- P5P4: 2.8 tons annually? That might be a wrong number.

- P5P4: ..."the AMOUNT of" landfilled waste...?

- Part 2.1. title: capital letter is not necessary in the word Criteria

- Table 1. Socio-Economic – CC: In the case of this subcriteria, i'm not sure that the higher value is surely better. This may even be detrimental in certain circumstances/conditions.

- Table 2: What is "solids" chamber? Is it a commonly used expression for this purpose? I did not know that. The further part is understandable of course.

- at least two of the four WTE technologies are not a solution to garbage disposal.

- how were the experts involved in the evaluation selected? Based on what methodology?

- P6 last paragraph: dot needed.

- Please centralize equation 2.

- Figure 4 is inaccurately edited in many places. There is also an error between the colors (e.g. Sophistication of technology, green line instead of orange). Please corrent the figure.

- 3. title: … Discussions: plural is not needed.

- Part 3.1. ... among other processes... (plural needed)

- Part 3.1.: A useful reference for supplementing: Vershinina, K., Dorokhov, V., Romanov, D., Nyashina, G., & Kuznetsov, G. (2020). Multi-criteria efficiency analysis of using waste-based fuel mixtures in the power industries of China, Japan, and Russia. Applied Sciences, 10(7), 2460.

- Table 6.: 1st column, 2dn row: Please add ... "and health" to the title as it is in the text.

- Please indicate and explain the calculation of the different priority vectors in table 6, as well as the other values except for the same categories.

- Cradle to grave: Please use quotation marks.

- P10: Emissioions: Spelling error.

- Fig. 5.: What does mergeformat mean and why is it here and elsewhere?

- P11P2: For me, this part needs some information regarding the energy balance and characteristics of the WTE technologies examined. That could confirm the statement of the experts in a practical way.

- Table 7: These indicators sounds me too theoretical. Practical information or maybe supplementation with different citations and research results could confirm the opinion of experts, connected to CAPEX and OPEX of these technologies.

- P12 last paragraph: error in word order; 3.15 is electric energy?; 901 MW: I think the measure could be MWh here.

- Legend under Fig. 6.: POH, not POC; CCI: This abbreviation is different as in the figure..; OMC: this is also different, please unify; JC/CJ: the letters are changed in the figure.

- In figure 8., the text is not visible.

- P16 …operate a four…: a is unnecessary

- The format of the reference list is only partially follows the instructions that is available among the Instructions for Authors document.

Author Response

Reviewer 3

Comments and Suggestions for Authors

General comments:

This study reports on evaluation of different waste to energy options in the Moscow region, using multi criteria analytical hierarchy process.

The manuscript is not really well written, however, the research is interesting. The structure follows the required structure, but the manuscript contains dozens of spelling and stylistic errors as well as shortcomings connected to the format and editing details required in the authors’ guide. Almost all parts of the manuscript should be improved. Furthermore, I have several technical questions regarding the details, which could be clarified and many supplements should be done. In my opinion, the manuscript's content is oversimplified regarding the WTE technologies and extensive English editing in needed

I believe that the four energy production methods are too few to make an informed decision, and that the individual energy production options are almost completely different, and in many cases it would be advisable to build systems containing two or more of them. Within the four main solutions, it would be necessary to share additional types (subcategories) to select the appropriate technology.

All in all, I suggest rejecting the manuscript due to the shortcomings of it. On the other hand, the author’s research is valuable, that is why I encourage the Authors to resubmit the article after modifications. In order to further develop it, I make the following recommendations.  

Note: I think, although this scientific method is also useful, an informed decision could be made using other analyzes and methods. I know that multi-criteria decision analysis is a commonly used method, but in my opinion it is quite subjective and theoretical. E.g. feasibility studies, scenario analyzes and LCA could be more useful in order to make an informed decision. Along with this, the analysis carried out is interesting, however, in my view it is less applicable to sound decision-making.

Technical recommendations:

1.- Probably the official template was not used during editing the manuscript. There are lots of shortcomings and inaccuracies regarding the format (no line numbers, that is why I cannot indicate the exact line numbers in my review report; different format in the Abstract, tables, references etc.).

In writing up the manuscript, the authors tried to follow up the journal guidelines for authors. However, it seems that there has been some changes occurred in the manuscript structure due to different versions of the Microsoft word used. To avoid such changes, the revised manuscript has been prepared in both word and pdf formats.

2.- Abstract: Landfill biogas: unnecessary capital letter.

The word landfill in the abstract has been written using small letter.

  1. - Page 2 (P2) Paragraph 1 (P1) (P2P1): World Bank report: this citation (2011) is too old, out-of-date. Please replace this and similar references and data with up-to-date sources and data.; tones: spelling error.

A new world bank report of 2018 has been used instead of 2012. The text and the reference list has been amended accordingly.

Kaza S., Yao L., Perinaz Bhada-Tata P. B. and Van Woerden F. (2018) What a Waste 2.0, A Global Snapshot of Solid Waste Management to 2050, World Bank, Washington, DC. USA, https://openknowledge.worldbank.org/handle/10986/30317

  1. - P2P1: No dot at the end of the sentence.

A dot has been added

5.- P2P2: WTE: Please insert the abbreviation only at the 1st appearance.

The full name has been deleted. Instead abbreviation WTE has been used.

6.- unnecessary parentheses (e.g. P2P2 reference 14, 18)

Unnecessary parenthesis has been deleted as requested.

7.- Figure 1: the given citation could be indicated below the title of the figure in a separate row.

Done

8.- What does the curved arrow mean between Recycling and Waste disposal? Why is it there, not below the last option?

This means that some of the nonrecyclable items find its way to the disposal site.

  1. - P2 last paragraph: Finding…criterion: This sentence is a bit too simple wording in such a scientific article, please rewrite it.

The sentence has been rewritten to be read as follows "Finding an optimal WTE technology is a challenging task that cannot be made based on a single criterion"

  1. - P2 last paragraph: decisions: is plural needed?

No plural not needed, already written as a single

  1. - P3P1: What is MSW? Municipal solid waste? Please indicate. This abbreviation is not yet explained.

Full name as Municipal solid waste has been written before the abbreviation.

  1. - Part 1.1. first sentence: unnecessary spaces.

Space has been deleted

  1. - P3 part 1.1. please unify the writing of million (capital is not needed).

It has been rewritten as million

  1. - dot is unnecessary before reference 27.

Dot has been deleted

  1. - Figure 2.: is it own construction? If not, please add the source here and elsewhere to other figures and tables.

Yes, figure 2 has been developed by the authors. This has been indicated in the figure caption.

  1. - P3 part 1.2.: missing dot at the end of the sentence and missing comma after ref. 28. Please check the whole document for missing characters.

Dot has been added and comma added.

  1. - P4. ref. [5.6.7]: the format is not as required.

Dots have been deleted and commas added

  1. - more than 500 m, but not more than 900 m from the residential areas? (500-900 meters away)

It has been modified as requested

  1. - P4 P2: duplication of words: that that, please remove one of them.

One of the words that has been deleted

  1. - P4P2: I recommend to use "based" instead of "depending".

The word depending has been replaced with based.

  1. - Figure 3.: - unnecessary "h" in the word Hhygienic; and I recommend to use "in" instead of "by"

Word Hygienic been corrected and by replaced with in as recommended.

  1. - I think GWh could be a better measure than Gcal, but the change is only an option.

Gcal is taken as is from the reference. Authors recommend to keep it as it is appeared in the reference.

  1. - P4 last paragraph, 42 million tons and 65 million: Is it an own calculation or not? Maybe from ref. 38.? What is the measure of the second one?

No it is not own calculations, it is reported by reference 38. The unit for 65 million is added as tons.

  1. - P5P1: The heating system's CO2 emissions"...; and Mt is for million or metric? Please clarify.

It is metric ton.

  1. - P5P2: "This ..., in particular": the values in this sentence is completely contradictory, please modify.

The sentence has been modified.

  1. - Please unify the use of Gcalorie in the text.

Already unified to Gcalorie

  1. - P5P4: 2.8 tons annually? That might be a wrong number.

Yes, wrong. It is 2.8 million tons rather than 2.8 tons. It has been corrected accordingly.

  1. - P5P4: ..."the AMOUNT of" landfilled waste...?

Yes, the word amount has been added to the statement.

  1. - Part 2.1. title: capital letter is not necessary in the word Criteria

Criteria has been changed to criteria

  1. - Table 1. Socio-Economic – CC: In the case of this subcriteria, i'm not sure that the higher value is surely better. This may even be detrimental in certain circumstances/conditions.

It is not stated here that the higher value of investment cost is better. It is rather talking about a technology that has the least initial cost.

  1. - Table 2: What is "solids" chamber? Is it a commonly used expression for this purpose? I did not know that. The further part is understandable of course.

It is a common terminology used in incineration plants, where incinerator has dual chambers, one for burning the solids and the second for burning the gases.

  1. - at least two of the four WTE technologies are not a solution to garbage disposal.

The four WTE technologies are used for the treatment of the full garbage stream or some components of it. Landfills can receive the whole stream, while the incineration and refuse derived fuel need to segregate the materials with high calorific value.

33 - how were the experts involved in the evaluation selected? Based on what methodology?

Sixteen experts and stakeholders with diverse administrative, technical and economic backgrounds and having expertise in the peer review process, who are familiar with the waste and energy sectors of Moscow region were interviewed and consulted.  A brainstorming sessions conducted. During the session, the objectives of the study were explained and the questionnaire was presented. After that the questionnaire was distributed to them and they were guided on how to carry the pairwise comparison process and to give the weights for various criteria and subcriteria.

  1. Work experience and correspondence of education to this profile

34 - P6 last paragraph: dot needed.

Dot has been added

35 - Please centralize equation 2.

Equation has been centralized.

  1. - Figure 4 is inaccurately edited in many places. There is also an error between the colors (e.g. Sophistication of technology, green line instead of orange). Please corrent the figure.

This distortion in the figure may be caused by having a different word version. The figure has been corrected accordingly.

  1. - 3. title: … Discussions: plural is not needed.

Discussions corrected to discussion.

  1. - Part 3.1. ... among other processes... (plural needed)

Corrected to processes

  1. - Part 3.1.: A useful reference for supplementing: Vershinina, K., Dorokhov, V., Romanov, D., Nyashina, G., & Kuznetsov, G. (2020). Multi-criteria efficiency analysis of using waste-based fuel mixtures in the power industries of China, Japan, and Russia. Applied Sciences, 10(7), 2460.

The recommended reference has been cited and added to the list of references.

Vershinina et al. (2020) used AHP among other multicriteria methods in efficiency analysis of using waste-based fuel mixtures in the power industries of China, Japan and Russia [50].

40 - Table 6.: 1st column, 2dn row: Please add ... "and health" to the title as it is in the text.

The word Health has been added as recommended

  1. - Please indicate and explain the calculation of the different priority vectors in table 6, as well as the other values except for the same categories.

The following paragraph has been added to explain the findings of the pair wise comparison in table 6.

It can be observed that the pair wise comparison revealed that environmental criterion has a demonstrated strong importance with respect to technical as well as with respect to socio-economic criteria. On the other hand, pair wise comparison showed that technical criterion is almost of equal to weak importance as compared to socio-economic criterion

  1. - Cradle to grave: Please use quotation marks.

A quotation mark has been used.

  1. - P10: Emissioions: Spelling error.

Spelling has been corrected.

  1. - Fig. 5.: What does mergeformat mean and why is it here and elsewhere?

Cannot get what the reviewer means by this comment.

  1. - P11P2: For me, this part needs some information regarding the energy balance and characteristics of the WTE technologies examined. That could confirm the statement of the experts in a practical way.

WTE technologies have been prioritized and ranked by the experts based on four main criteria and 9 subcriteria. The characteristics of the technologies are described in table 2.

  1. - Table 7: These indicators sounds me too theoretical. Practical information or maybe supplementation with different citations and research results could confirm the opinion of experts, connected to CAPEX and OPEX of these technologies.

The authors tried to confirm their research findings by comparing them with the findings of other researchers who conducted similar studies. Furthermore, the authors tried to explain their results based on the status of the Russian solid waste and energy sectors. The following statements  extracted from the manuscript are examples of such comparison.

 Khoshand et al (2018) [25] also found that the occupational health and safety has the highest priority as compared to other criteria.

This indicates that regardless of the technology used and its complexity, the main issue in selecting that technology is the amount of the generated energy from that WTE technology. Qazi et al. (2018) [52] found that energy production is also of highest priority.

  1. - P12 last paragraph: error in word order; 3.15 is electric energy?; 901 MW: I think the measure could be MWh here.

The wording has been corrected and the word electric has been added. We think that since it is a power potential rather than energetic potential, the unit should remain MW not MWh.

  1. - Legend under Fig. 6.: POH, not POC; CCI: This abbreviation is different as in the figure..; OMC: this is also different, please unify; JC/CJ: the letters are changed in the figure.

POC has been changed to POH, but CCI is the same on the figure and in the text below the figure. OMC and CJ has been changed as per the reviewer request

  1. - In figure 8., the text is not visible.

The text on figure 8 has been increased in order to be visible.

  1. - P16 …operate a four…: a is unnecessary

operate a four has been deleted as requested by the reviewer

  1. - The format of the reference list is only partially follows the instructions that is available among the Instructions for Authors document.

Efforts has been made to comply with the journal style in listing the references.

Round 2

Reviewer 3 Report

I recommend to indicate the content of the Authors’ reply (they have given for question 32), in the conclusion part (only the content, not the same text): The four WTE technologies are used for the treatment of the full garbage stream or some components of it. Landfills can receive the whole stream, while the incineration and refuse derived fuel need to segregate the materials with high calorific value.

The content is acceptable now, however, I still think that these “solutions” or technologies are difficult to compare with each other. Furthermore, I believe that the analysis performed has little practical benefit/significance compared to other possible evaluation methods. On the other hand, the basic methodology is appropriate.

The manuscript still contains several editing errors (unnecessary characters and formatting errors), and there are no line numbers in the PDF document.

Author Response

RE:  " Using Multi-Criteria Decision Analysis to Select Waste to Energy Technology for a Mega City: The Case of Moscow, Ref.: Manuscript ID: sustainability-978985

 Dear Editor,

We appreciate the time that has been taken by the reviewer number 3 to give more comments and suggestions in the second review cycle. We have addressed all of the concerns raised by the reviewer in a very thoughtful and careful manner.  We hope that we have responded favorably to each of the concerns presented.

Note:  The revised submission includes: Response to reviewer comments and Revised manuscript (changes marked with red font).

Kind regards,

Dr. Anna Kurbatova , Corresponding Author

Reviewer # 3

Comments and Suggestions for Authors

  1. I recommend to indicate the content of the Authors’ reply (they have given for question 32), in the conclusion part (only the content, not the same text): The four WTE technologies are used for the treatment of the full garbage stream or some components of it. Landfills can receive the whole stream, while the incineration and refuse derived fuel need to segregate the materials with high calorific value.

The conclusion section has been updated as recommended by the reviewer. The following statement has been added to the section which is highlighted in red font.

AHP analysis was used to make a decision among four WTE alternatives which treat the whole stream of solid waste or some of its components.

  1. The content is acceptable now, however, I still think that these “solutions” or technologies are difficult to compare with each other. Furthermore, I believe that the analysis performed has little practical benefit/significance compared to other possible evaluation methods. On the other hand, the basic methodology is appropriate.

Many published research articles and studies have carried such a comparison between WTE technologies. The current article mentioned such publications, please see the references list.

Regarding the evaluation methods, there are several decision making tools, each has its advantages and limitations. As for the solid waste management, AHP is a widely used tool for decision making process. By reviewing 279 articles published between 1980 and 2015, Coelho et al. (2017) reported that AHP was the dominant decision making process used among other processes with 32% (91 articles) of the total. This reference has been added to the references list as reference no. 47, where the number of references has been updated accordingly.

Lanshina T. A., Laitner J. A., Potshinkov V. Y. and Barinova V. A. (2018) The slow expansion of renewable energy in Russia: Competitiveness and regulation issues, Energy Policy,120, pp 600-609, https://doi.org/10.1016/j.enpol.2018.05.052

  1. The manuscript still contains several editing errors (unnecessary characters and formatting errors), and there are no line numbers in the PDF document.

The manuscript has been scanned for editing errors and corrected where possible. The word version of the manuscript has been updated to add the line numbering as requested by the reviewer.
